# Homologous Recombination Deficiencies and Hereditary Tumors

**DOI:** 10.3390/ijms23010348

**Published:** 2021-12-29

**Authors:** Hideki Yamamoto, Akira Hirasawa

**Affiliations:** Department of Clinical Genomic Medicine, Graduate School of Medicine, Dentistry and Pharmaceutical Sciences, Okayama University, 2-5-1 Shikata-cho, Kita-ku, Okayama 700-8558, Japan; hir-aki45@okayama-u.ac.jp

**Keywords:** homologous recombination deficiency (HRD), hereditary tumor, germline, cancer predisposition, multi-gene panel testing (MGPT), BRCAness

## Abstract

Homologous recombination (HR) is a vital process for repairing DNA double-strand breaks. Germline variants in the HR pathway, comprising at least 10 genes, such as *BRCA1*, *BRCA2*, *ATM*, *BARD1*, *BRIP1*, *CHEK2*, *NBS1*(*NBN*), *PALB2*, *RAD51C*, and *RAD51D*, lead to inherited susceptibility to specific types of cancers, including those of the breast, ovaries, prostate, and pancreas. The penetrance of germline pathogenic variants of each gene varies, whereas all their associated protein products are indispensable for maintaining a high-fidelity DNA repair system by HR. The present review summarizes the basic molecular mechanisms and components that collectively play a role in maintaining genomic integrity against DNA double-strand damage and their clinical implications on each type of hereditary tumor.

## 1. Introduction

Homologous recombination (HR) is a tightly regulated molecular mechanism essential for maintaining genomic integrity against DNA double-strand breaks (DSBs) in a wide range of species, including yeast and humans. Compared to other known mechanisms, such as non-homologous end-joining (NHEJ) and microhomology-mediated end-joining (MMEJ), the most noteworthy characteristics of HR for DSB repair are error-free procedures and outcomes using sister chromatids and highly conserved molecular associates [1,2]. The representative genes involved in HR are *BRCA1* and *BRCA2*, whose deleterious variants are associated with hereditary breast and ovarian cancer (HBOC) syndrome [3]. Hereditary tumors, including HBOC, are characterized by constitutional variants which lead to intrafamily accumulation of specific types of cancer, early age onset, and/or multiple tumor development in a synchronous or metachronous manner. Inherited predispositions to cancer, sarcoma, or even benign tumors are mainly due to germline loss-of function in cell cycle regulation or DNA damage repair [4].

For error-free DSB repairing, HR can be considered the most important, or perhaps even the only mechanism. Recent clinical trials of molecular-based precision medicine on patients and their families (i.e., probands and relatives) suffering from hereditary tumors have yielded encouraging evidence for utilizing molecular-based knowledge of HR not only against tumor development associated with *BRCA1* and *BRCA2*, but also for other HR-related gene variants carrying moderate or low tumor development risks [5,6]. In this work, we provide an overview of the mechanisms and essential molecular components involved in an efficient HR with their impacts on hereditary tumors. We also discuss the clinical utility of multi-gene panel testing (MGPT) for the diagnosis of HR-related hereditary tumors as well as for therapeutic and preventive strategies against them.

## 2. Molecular Components and Cascade Overview of HR

HR repair is an essential molecular mechanism for maintaining genomic stability by repairing double-strand DNA breaks (DSBs) or lesions that stall DNA replication forks [7]. DSBs are potentially more complex and difficult to repair than other types of DNA damage. DSBs can be caused not only by exogenous sources, such as ionizing radiation and toxic chemicals, but also by endogenous genotoxins, including reactive oxygen species and metabolic byproducts. High-fidelity and timely repair of DSBs is necessary for maintaining genomic integrity and cell survival by minimizing mutagenesis or improper apoptosis, which may otherwise lead to carcinogenesis [8,9]. HR is a highly conserved mechanism in eukaryotes, from yeast to humans. HR in budding yeast *Saccharomyces cerevisiae*, for example, is catalyzed by proteins encoded by *RAD52* epistasis group genes such as *RAD50*, *RAD51*, *RAD54*, *RAD55*, *RAD57*, *RAD59*, *XRS2* (*NBS1* or *NBN* in humans), and *MRE11*, including *RFA1–3*, the heterotrimeric complex of RPA [10]. There is a conserved appearance in the foci formation with DNA-damage-related products at the site of DNA damage [11].

In mammalian cells, the macro-complex consists of BRCA1 and BARD1, which bind through the RING finger domain at the N-terminus; components such as phosphorylated ABRA1 binding through the BRCT domain at the carboxyl-terminus; and NBA1, RAP80, etc. All these components are recruited to the sites of DNA damage along with BRCA1 [12,13]. These macro-complex structures are observed through DNA damage checkpoint activation, which leads to G2-M checkpoint arrest. HR uses undamaged sister chromatids as templates to enable precise repair and functions during the cell cycle phases of late S to G2. RAD51 recombinase, which is the final effector of the HR cascade, and is recruited to the sites of DNA damage as a consequence of the interaction of BRCA1 and BRCA2 through PALB2, which is a key triplet complex of HR (Figure 1, Part 1). RAD51 is carried by binding to BRCA2 through eight conserved motifs of BRC repeats in the middle of the molecule and at the unrelated carboxyl-terminus region of the TR2 domain [14]. Once RAD51 is loaded and released on the end-resected DNA damage site, RAD51 forms a nucleofilament and leads to strand invasion. Repair proceeds from the invading DNA end to the second end, which is captured by two Holliday junctions and is thereafter resolved as a crossover or dissolved in a non-crossover form [15].

### 2.1. Key Initial Step for HR: Sensing DSBs

To proceed to HR, a key initial step is the sensing of the damaged sites by the MRE11–RAD50–NBS1 complex (MRN complex) [16,17]. The molecular function of MRE11 was originally identified as a 3′ to 5′ exo- and endonuclease, whereas RAD50 was found to be an ATPase [18,19]. Extensive studies have revealed that the functions of MRN complex are categorized in the following three roles. Firstly, it holds the damaged ends of DNA in DSB lesions structurally by RAD50. Secondly, it creates single-stranded DNA (ssDNA) by nucleolytic processing to expose 3′ of DNA, that is, 3′ overhangs by the end resection of the 5′ to 3′ strand for which MRE11 is functional alongside BRCA1 [8]. Both sides of the DSBs are immediately acted upon by human replication protein A (RPA), an ssDNA-binding protein, to melt DNA secondary structures and to prepare for RAD51 nucleoprotein filament formation. Lastly, NBS1 (alternative name: NBN), a substrate of ataxia telangiectasia-mutated (ATM), is responsible for the recruitment and checkpoint activation of ATM.

ATM is traditionally known as a multifunctional serine/threonine kinase, which stabilizes and activates the tumor suppressor protein p53. Upon DNA damage, ATM is recruited to the damaged site and leads to subsequent check-point activation during the G1/S checkpoint [7,20]. ATM and ATM-Rad3-related (ATR) phosphorylate multiple downstream targets such as p53, H2AX, and BRCA1 either directly or via CHEK2 gene coding checkpoint kinase 2 protein (CHK2) activation [21]. BRCA1 is assisted by BRCA1-associated RING domain 1 (BARD1) and BRCA1 interacting helicase 1 (BRIP1), which acts as a scaffolding protein that organizes the assembly with other repair proteins (Figure 1, Part 2). BRCA1 interacts with BARD1 via the RING-finger domain near the N-terminus of BRCA1, both of which encode a nuclear export signal (NES). BRCA1 forms three different types of complexes exclusively with phosphorylated Abraxas (ABRA1), BRCA1-associated C-terminal helicase 1 (BACH1/FANCJ/BRIP1), or CtBP-interacting protein (CtIP) through the BRCT domain near the C-terminus of BRCA1 [22,23]. As a consequence of interacting with these different proteins, BRCA1 plays pleotropic roles, including DNA damage resistance, ubiquitination, gene transcription, and cell cycle progression, such as G2-M checkpoint control [24] (detailed information is found in *Molecular Diagnosis and Targeting for Gynecologic Malignancy*, ISBN 978-981-33-6013-6 (eBook)) [25].

Under the mediation of PALB2, a linker protein between BRCA1 and BRCA2, RAD51 is recruited to the DSB sites by binding to BRCA2. Once successfully released from BRCA2, RAD51 forms nucleofilaments and allows DNA to invade the homologous double helix. HR progresses by DNA synthesis using homologous DNA as a template, leading to second-end capture and double Holliday junction formation, ultimately ending in either resolution or dissolution.

### 2.2. ATM, CHEK2, BRIP1, and BARD1: Activators and Partners of BRCA-1 and Their Roles in Cancer Susceptibility

*ATM* and *CHEK2* are recognized as the most common breast cancer susceptibility genes, and their attenuation is associated with BRCAness, namely phenocopies of BRCA1/2 defects. In a single institutional review of 1185 breast cancer cases using multi-gene panel testing (MGPT), a high occurrence of bilateral tumors was observed in 26.3% of the *ATM* germline variant carriers and in as much as 41.2% of *CHEK2* variant carriers [26]. Histological analyses of breast cancer subtypes showed that there were varying trends between *ATM* and *CHEK2*. Over half (64.3%) of *CHEK2*-associated tumors were luminal A-like subtype (enriched for hormonal receptors, less proliferative, and low-grade), whereas nearly half (56.2%) of *ATM*-associated tumors were luminal B-like subtype (low expression of hormonal receptors, but more proliferative and high-grade) and/or human epidermal growth factor (HER2)-negative type. Lobular carcinoma was observed in 21.4% of *CHEK2*-related invasive tumors. A quarter of ATM-related breast cancer and approximately one-third (30%) of *CHEK2*-related breast cancers were in situ carcinomas [26].

ATM proteins are multi-functional. They are conventionally found as an activating factor of p53 in response to DNA damage and phosphorylate BRCA1 following irradiation injury [27]. Phosphorylation of BRCA1 by ATM is an important procedure for G2/M and S-phase checkpoint activation, in which BRCA1 plays a role [28]. CHK2, protein product of *CHEK2*, is also categorized as a serine/threonine kinase that is activated by ATM-mediated phosphorylation of SQ/TQ sites (cluster motifs containing seven serine (S) or threonine (T) residues followed by glutamine (Q)) in response to DNA damage [29]. Substrate spectra of CHK2 kinase are wide-ranging from cell cycle controls, such as phosphatases including Cdc25A, Cdc25C, and phosphoinositide 3-kinase (PI3K), to cell death signaling constituted by p53-MDM2 interplay, and DNA damage repair pathway through BRCA1. CHK2-mediated BRCA1 phosphorylation is important for the BRCA1–PALB2–BRCA2 effector complex functions in the HR pathway.

BRIP1, an alternative name for BRCA1-associated C-terminal helicase (BACH1) or Fanconi anemia subtype J (FANCJ), interacts with BRCA1 through the BRCT domains at the carboxyl-terminus (Figure 1, Part 2). BRIP1 was originally identified in the investigation of Fanconi anemia [30]. BRIP1 is physiologically essential for the maintenance of genomic integrity, removing DNA-bound proteins, stabilizing replication forks, and unwinding substitutive DNA structures along with RPA [31]. BRIP1 acts as a tumor suppressor through its interaction with BRCA1 [32]. Several truncating variants of *BRIP1* are conventionally known to be associated with breast cancer with low or moderate penetrance [33]. In recent years, it has been recognized that deleterious germline variants of *BRIP1* are highly susceptible to ovarian cancer [34]. There is an emerging report showing potential associations between germline pathogenic variants (GPVs) of *BRIP1* and colorectal cancer based on next-generation sequencing (NGS) analysis of multiple genes [35].

The BRCA1–BARD1 complex is essential for BRCA1 to function as an E3 ubiquitin ligase on a molecular basis [8,28]. GPVs in *BARD1* have been associated with early onset of breast cancer [36]. For the clinical management of GPV carriers of *ATM*, *CHEK2*, or *BARD1*, all of which are autosomal dominant, it is recommended to perform imaging screenings for preventive risk management against breast cancer. This may be done by annual mammograms with or without tomosynthesis and, by performing breast magnetic resonance imaging (MRI) starting at age 40, according to the National Comprehensive Cancer Network^®^ (NCCN) Guidelines^®^ Version 1.2022 as of August 2021 (www.nccn.org, accessed on 19 November 2021). Breast cancer risk in females with GVPs of *BRIP1* is stated as “potential”, with insufficient evidence for risk management whereas there are increased risks for ovarian cancer, justifying the consideration of risk-reducing salpingo-oophorectomy (RRSO) at around 45–50 years of age (with a discussion) or earlier, based on specific family history of an earlier onset of ovarian cancer (NCCN Guidelines^®^ Version 1.2022, www.nccn.org, accessed on 19 November 2021).

### 2.3. PALB2: Bridging Mediator of BRCA1 and BRCA2 and the Roles in Cancer Susceptibility

Partner and localizer of BRCA2 (PALB2) was discovered as an interacting protein with BRCA2 [37]. PALB2 mediates complex formation together with BRCA1 and BRCA2 and promotes BRCA2 recruitment to DSB sites. In addition to these functions, *PALB2* was identified as an associated gene for Fanconi amenia (subtype N, FANCN) and pediatric malignancies such as Wilms’ tumor or medulloblastoma. It is also implicated in the susceptibility to breast cancer in adults, conferring a 2.3-fold higher risk of cancer development [38]. However, in later studies, penetrance of variants of *PALB2* in breast cancer development was found to be distinctively varied in different populations [39]. Breast cancer risk due to loss-of-function variants of *PALB2* overlapped with that of *BRCA2* variants, validating the direct interaction of PALB2 with BRCA2 [40]. In recent years, the spectrum of clinical preventive risk management for carriers with GPVs of *PALB2* has been extended to epithelial ovarian cancer and pancreatic cancer, with absolute risks at 3–5% and 5–10%, respectively, according to NCCN Guidelines^®^ Version1.2022 (www.nccn.org, accessed on 19 November 2021).

There is also a suggestive case report showing a potential involvement of GPVs of *PALB2* and *NBS1* (*NBN*) in the susceptibility to pancreatic cancer [41]. For presumed high-risk individuals with a pronounced history (e.g., at least two affected cases within first-degree relatives) of familial pancreatic cancer, it is suggested to conduct routine follow-ups with endoscopic ultrasound (EUS) and MRI or magnetic resonance cholangiopancreatography (MRCP) as surveillance of pancreatic cancer, in accordance with the International Cancer of Pancreas Screening (CAPS) established in 2013 by Johns Hopkins University [42] and with the latest version of the NCCN Guidelines^®^ [43].

### 2.4. RAD51, RAD51C, RAD51D: The Final Effector of HR and Paralogs

Recombinase RAD51 is a critically important molecule, as it is the final effector in the HR cascade. RAD51 is recruited to the DSB sites after being carried under the mediation of BRCA2. RAD51 binds to BRCA2 via the eight conserved short motifs of BRC repeats and a carboxyl-terminus region called the TR2 domain (Figure 1, Part 3). Deleterious variants of *BRCA2* at BRC repeats can, therefore, compromise the interaction of BRCA2 and RAD51, potentially causing HBOC phenotypes [44]. Binding of RAD51 to BRCA2 at the carboxyl terminus is known to be dependent on the cyclin-dependent kinase 2 (CDK2)-mediated phosphorylation status of BRCA2 [45]. After loading RAD51 successfully on the resected DNA ends, RAD51-bound DNA forms nucleofilaments and invades the homologous double helix.

DNA polymerase synthesizes new DNA structures, using homologous DNA as a template with ssDNA as a primer. RAD51C (RAD51L2) and RAD51D (RAD51L3) constitute RAD51 paralogues, with other members such as RAD51B (RAD51L1), XRCC2, and XRCC3. RAD51C and RAD51D form a subcomplex with RAD51B and XRCC2 (BCDX2 complex) (Figure 1, Part 4). They assist the recruitment of RAD51 under the regulation of BRCA1–PALB2–BRCA2 effector complex [28]. GPVs of *RAD51C* and *RAD51D* are known to impart susceptibility to breast cancer at absolute risks of 15–40% (NCCN Guidelines^®^ Version1.2022 at www.nccn.org, accessed on 19 November 2021), while both of them are shown to moderately increase susceptibility to ovarian cancer, such as a six-fold higher risks for carriers of *RAD51D* variant [46]. The penetrance of *RAD51D* for ovarian cancer is at moderate risk, which is almost consistent across ethnicities [47]. In the NCCN Guidelines^®^ Version 1.2022, it is stated that there are not sufficient data so far to suggest preventive management of breast cancer in carriers of GPVs of *RAD51C* or *RAD51D* (*RAD51C*/*D*). Conversely, RRSO is considered at age of 45–50 years for the clinical management of ovarian cancer associated with *RAD51C*/*D* variants (www.nccn.org, accessed on 19 November 2021). There is also new evidence regarding the clinical significance of *RAD51D* splicing variants that can be detected in various types of tumors, including non-epithelial sarcoma [48].

## 3. HR-Based Precision Oncology for Hereditary Tumor Clinics

A lot of clinical evidence has been accumulated concerning HR, the impairment of which is associated with inherited susceptibility to hereditary tumors. A representative hereditary tumor due to germline deficiency of HR, which is represented by alterations in *BRCA1* or *BRCA2* (*BRCA1*/*2*), is the hereditary breast and ovarian cancer (HBOC) syndrome [49]. Pathogenic attenuations in either *BRCA1* or *BRCA2* account for 25–40% of familial breast cancers and up to 10% of all breast cancers [50,51]. In several large-scale case-control studies of familial cancer patients and whole exome germline sequencing, alternative candidate genes conferring susceptibility at various intensities have been identified in breast cancer and in other HBOC-related tumors such as ovarian, pancreatic, and prostate cancers (Table 1).

According to a sequencing study conducted by Welsh et al. for 360 cases of primary ovarian, peritoneal, or fallopian tube carcinomas, irrespective of the family history, 24% of the cases carried germline loss-of-function variants of either *BRCA1*, *BRCA2*, or other genes, including those involved in HR pathways such as *CHEK2*, *MRE11A*, *NBS1* (*NBN*), *RAD50*, *RAD51C*, or in the FA pathway such as *BRIP1*/*FANCJ*, *PALB2*/*FANCN*, and *BARD1*. The rest of the population carried germline variants in *TP53*, a tumor suppressive DNA damage response gene, or *MSH6*, a mismatch-repair gene. The proportion of germline variants in any of the genes other than *BRCA1*/*2* accounted for 6% of the total cases examined, which, in turn, accounted for one-third of the patients with germline variants in their study [82]. In tumors with germline alterations of HR-related genes, there are emerging new concepts termed as BRCAness, phenocopies of *BRCA1*/*2* defects, expressing communal phenotypes such as the organs in which malignancy develops and the potential clinical responses to platinum-based therapies or vulnerability to poly(ADP-ribose) polymerase inhibitors (PARPi) [83,84,85]. With respect to the penetrance in each type of cancer, *BRCA1*/*2* is known to be the strongest factor responsible and therefore, is categorized as a high-risk genetic factor in imparting susceptibility to specific types of cancer.

According to the results of an international *BRCA1*/*2* cohort study, the cumulative risk of developing breast cancer among *BRCA1*/*2* variant carriers reaches approximately 70% by the age of 80 (72% in *BRCA1* variant carriers and 69% in *BRCA2* variant carriers) [60]. For ovarian cancer, the risk is 17% and 44% for *BRCA1* and *BRCA2* variant carriers, respectively, by the age of 80. The estimated frequencies of developing malignancies are 0.1% for prostate cancer and 0.5% for pancreatic cancer, while the relative risks for their lifetime incidence were reported to increase significantly (up to 20-fold for prostate cancer and 10-fold for pancreatic cancer) upon association with *BRCA2* [28,86]. According to a phase II study conducted by Tug et al., germline variants of *PALB2* are presumed to have a favorable response to treatment with olaparib, a PARPi, in metastatic breast cancer patients [87]. The overall survival of pancreatic adenocarcinoma patients with germline HR-related gene variants, such as *ATM* alterations, is significantly longer compared to that of non-variant carriers [88]. This evidence offers the validity of routine sequencing of not only *BRCA1*/*2* but also of the wide range of HR-related genes as part of biological characterization and for treatment decisions using PARPi, at least in the cases of breast and pancreatic cancer.

### 3.1. HR Deficiency (HRD) and HRD-Associated Genes

HR deficiency (HRD) is most prevalent in ovarian, breast, prostate, and pancreatic cancers. In a pan-cancer cohort reported by Nguyen et al., the HRD frequency in ovarian and breast cancers is followed by that in pancreatic and prostate cancers. The prevalence of HRD in patients with any of four types of cancer was reported to be up to 85% by an analysis in which HRD was predicted through specific single nucleotide variants (SNVs), insertions/deletions (indels), structural variants (SVs), or loss-of-heterozygosity (LOH) [89]. Assessing HRD status by the region of LOH has been proposed as a useful indicator for defective *BRCA1*/*2*, at least in ovarian cancer [90]. In a systematic review and meta-analysis of the prevalence of HRD in both somatic and germline contexts among patients with pancreatic cancer, HRD descriptions based on genomic scarring and surrogate markers such as point mutational and structural variants were well-characterized with HR-related genes such as *BRCA1*, *BRCA2*, *PALB2*, *ATM*, *ATR*, *CHEK2*, *RAD51*, and the *FANC* genes [91]. Their study showed that HRD prevalence in pancreatic cancer ranged from 14.5–16.5% through targeted NGS and was as high as 24–44% by whole-genome or whole-exome sequencing.

It has been recognized that HRD is associated with cellular sensitivity of ovarian cancer to crosslinking agents such as platinum salts and topoisomerase inhibitors [92]. In contrast, according to the American Society of Clinical Oncology (ASCO) guidelines published in 2020, current HRD assays do not provide significant differentiation of patient response to PARPi to routinely recommend their use [93]. These controversial discussions regarding the clinical significance of HRD in the responsiveness to PARPi might be due to different pipeline methodologies or criteria for HRD evaluation. MyChoice test (Myriad Genetics, Salt Lake City, UT, USA), for example, determines HRD status solely in tumor tissues through the combined analyses of pathogenic variants in *BRCA1* and *BRCA2* based on sequencing and large rearrangements, and by assessing genomic instability in three biomarkers: LOH, telomeric allelic imbalance (TAI), and large-scale state transition (LST) (https://myriad.com/, accessed on 23 November 2021). Another weighted model, HRDetect, was developed as a tool to predict *BRCA1*/*2* deficiency in tumors that reflects selective sensitivity to PARPi. HRDetect covers both germline and somatic events by integrating all classes of mutational signatures in the analyses [94].

Among PARPi (e.g., niraparib, olaparib, and rucaparib), olaparib was first approved by the US Food and Drug Administration (FDA) in 2018 for the treatment of patients with germline *BRCA*-mutated, HER2-negative metastatic breast cancer who were treated with chemotherapy (www.fda.gov, accessed on 17 November 2021) based on the OlympiAD trial [95]. Later in the same year, olaparib was approved for the maintenance of response in the first-line treatment of patients with deleterious or suspected deleterious germline or somatic *BRCA* mutated advanced epithelial ovarian, fallopian tube, or primary peritoneal cancer who respond completely or partially to platinum-based first-line chemotherapy. According to SOLO1 phase III trial, patients showed a reduced risk of progression or death with maintenance olaparib as compared to placebo by 70% [96]. Another phase III trial, the PAOLA-1 trial, showed that when maintenance olaparib was added to the first-line treatment with bevacizumab (a vascular endothelial growth factor (VEGF) monoclonal antibody), patients with advanced ovarian cancer presented significantly favorable outcomes as compared to bevacizumab plus placebo in the progression-free survival of patients with HRD-positive tumors, including those without a *BRCA* variant (HRD-positive and *BRCA* wild-type) [97]. In the VELIA trial, however, hazard ratios (HRs) of veliparib maintenance therapy in patients with newly diagnosed, high-grade serous ovarian carcinoma was similar in HRD-positive with *BRCA* wild-type (HR, 0.74; 95% CI, 0.52 to 1.06) and non-HRD tumors (HR, 0.81; 95% CI, 0.60 to 1.09) [98]. These somewhat controversial results are perhaps due to the different evaluation criteria for HRD using different cutoff values of genomic instability scores. The FDA has approved the indication criteria of niraparib for patients with advanced ovarian cancer, based on the cancer status determined as HRD-positive by HRD testing of BRCA mutated and/or a genomic instability score of over 42. Among HR effector genes potentially presumed as being associated with HRD status, apart from *BRCA1*/*2*, would be *ATM*, *BARD1*, *BRIP1*, *CHEK1*, *CHEK2*, *FANCA*, *FANCC*, *MRE11A*, *NBS1* (*NBN)*, *PALB2*, *RAD50*, *RAD51*, *RAD51B*, *RAD51C*, *RAD52*, and *RAD54L* [99]. In a pan-cancer study conducted by Nguyen et al., biallelic activation of *BRCA1*, *BRCA2*, *RAD51C*, and *PALB2* were reported as the most common genetic causes of HRD [89].

### 3.2. Differential Features of BRCA1-Type and BRCA2-Type

Considering the distinct molecular roles of BRCA1 and BRCA2, it is understandable that the pathogenic variants of *BRCA1* and *BRCA2* are associated with different subtypes in cancers. *BRCA1*-mutant breast tumors, for example, are typically negative for estrogen receptor alpha (ER alpha) [100]. A disease-specific microarray analysis of *BRCA1*-mutant breast tumor tissue over sporadic tumors expressing wild-type *BRCA1* found that the expression of *ESR1*, which is a coding gene for ER alpha, was significantly lower in *BRCA1*-mutant tumor tissue than in sporadic tumor tissues. In vitro analysis using breast cancer cell lines further identified a mechanism by which *ESR1* expression is positively regulated by direct binding of BRCA1 to the *ESR1* promoter, being recruited under the regulation of the transcription factor Oct-1. Clinically, *BRCA1* variant carriers have a higher risk of developing triple-negative (TN) status in breast cancer, which is negative for ER, progesterone receptor (PgR), and HER2 [101].

*BRCA2* variants typically cause ER-positive luminal subtypes, showing slow proliferation with low grade clinical aggression of breast cancer [32]. In a cohort study of patients with *BRCA* germline-mutated breast cancer, immunohistochemical analyses showed that the luminal A-like subtype, which is strongly positive for hormone receptors (ER and/or PgR) and low-proliferative and low-grade subtypes, were more frequently observed in *BRCA2*-mutated (35%) than in *BRCA1*-mutated (9%) [102].

According to a large-scale international joint consortium, it was shown that histopathological features of breast cancer refine the likelihood estimation for *BRCA1* or *BRCA2* variant status, which can be informative for genetic testing and clinical management of the patients involved [103]. Histologically, it is well-established that medullary breast carcinoma, which displays a basal-like molecular subtype, is associated with germline variants in *BRCA1* [104]. In prostate cancer, germline *BRCA2* variant carriers had worse clinical outcomes than non-variant carriers when treated with surgery or radiotherapy [105]. It has been shown that the *BRCA2*-mutant in prostate cancer might be associated with the histological characteristics of intraductal carcinoma of the prostate (IDC-P), and a tendency for higher incidence of biochemical relapse after surgery. In a pan-cancer HRD prediction study, it was found that biallelic inactivation of *BRCA1*, *BRCA2*, *RAD51C*, or *PALB2* is the most common genetic cause of HRD across cancer types, with the inactivation of the latter two genes, *RAD51C* and *PALB2*, resulting in the same mutational footprints of *BRCA2*, the so-called *BRCA2*-type HRD [89].

### 3.3. Reversion of BRCA1/2 Variants and Other Resistance Mechanisms to PARP Inhibitors or Platinum Chemotherapy

As somatic events in *BRCA1* or *BRCA2*, genomic reversion is known as a mechanism for acquiring resistance to PARPi or platinum chemotherapy in cancer cells [106,107]. Revertant *BRCA1*/*2* alleles due to secondary intragenic alteration ameliorates the wild-type *BRCA1*/*2* open reading frame. The reconstitution of functional BRCA1/2 protein restores intact HR and cells subsequently acquire resistance to PARPi or platinum-based chemotherapy. By comparing the analyses of tumor whole-exome sequencing before and after acquired resistance to PARPi or platinum chemotherapy, it has also been proposed that HR restoration is caused by increased DNA end resection due to *MRE11A* amplification [108]. It was also reported that no RAD51 foci were observed in pre-resistant tumors, while they were detected in post-resistance tumors, suggesting that RAD51 foci formation is an indicator of HR restoration and could be a potentially useful biomarker to predict responses to later DNA-damaging therapy. Biallelic inactivation of *TP53BP1* and a histone methyltransferase *KKMT2C*, or the reduced expression of *TP53BP1*, were also shown as genetic alterations essential for HR proficiency, and correlate with response and resistance to PARPi/platinum therapy.

## 4. Rational Strategies for Diagnosis of Hereditary Tumors Using Multi-Gene Panel Testing (MGPT)

The development of NGS technologies has paved the way for simultaneous analyses of multiple candidate cancer susceptibility genes through massive parallel sequencing. This recent trend in the research field is becoming widely common with large-scale non-biased or population-based analyses covering multiple genes. These strategies enable genetic characterization for the somatic and effective detection of the germline predisposition to each type of cancer for both probands and their relatives. Since a US Supreme Court Judgement invalidated the patent of Myriad Genetics on *BRCA1*/*2* genetics tests in 2013 [109], multi-gene panel testings (MGPTs) including *BRCA1*/*2* can be provided by other Clinical Laboratory Improvement Amendments (CLIA) or College of American Pathologists (CAP)-certified clinical laboratories for clinical use. MGPTs generally cover at least 10 common HR-related genes such as *ATM*, *BARD1*, *BRCA1*, *BRCA2*, *BRIP1*, *CHEK2*, *NBS1* (*NBN*), *PALB2*, *RAD51C*, and *RAD51D*, all of whose germline alterations are associated with BRCAness phenotypes for predisposition to breast, ovarian, prostate, or pancreatic cancer.

According to literature reviews performed by the authors, the detection of any of the GPVs is observed most frequently in ovarian cancer at 17.8–29%, followed by breast and pancreatic cancers at 5.8–9.5% and 6.7–9.3%, respectively, among all cancer patients involved in each type of cancer [47,52,82,110,111,112,113,114,115,116,117,118] (Figure 2). The frequency of GPVs observed in prostate cancer was significantly more in the USA (20.3%) than in Japan (3.0%), based on the results of MGPT analyses performed independently in the two countries. Deleterious variants of HR-related genes other than *BRCA1*/*2* were dominant in pancreatic cancer patients as well as prostate cancer patients in the USA, whereas *BRCA1*/*2* variants were most frequently detected as inherited predisposition factors in breast and ovarian cancers in the USA, China, and Japan.

There is also emerging discussion over the possible ethnic differences in germline variants of HR genes represented by *BRCA1* and *BRCA2* or *ATM*. The Global Alliance for Genomics and Health (GA4GH) launched a framework known as the BRCA Exchange (https://brcaexchange.org/, accessed on 20 December 2021) for international data sharing of pathogenic variants in *BRCA1* and *BRCA2*. Another international workshop for the *ATM* and cancer risk has begun to collect population-based worldwide data of germline variants in *ATM* associated with HBOC [119]. Through those international initiatives for data sharing, it is reasonably understood that ethnic background can be taken into account for the interpretation of variants of HR-related genes and risk intervention strategies for HBOC.

MGPT can be a pervasive modality for hereditary tumor diagnosis owing to its advantages of increased throughput and reduced overall turnaround time, duplication, and cost. Although the method potentially increases the chances of encountering variants of unknown significance (VUS), optimizing MGPT at an appropriate time point for each proband and relative will be essential in the clinical practice of hereditary tumors including HR.

## 5. Conclusions

We have reviewed and discussed functions of representative HR molecules that are highly conserved across species and are involved in repair mechanisms, which are complex and precise against DNA double-strand breaks. HR consists of four main steps: sensing, end resection, loading of RAD51, and strand invasion. The penetrance of each HR-related gene variant varies, reflecting the distinct risks and proper intensities of cancer susceptibility and incidence of hereditary tumors. Pathogenic variants in *BRCA1*/*2* convey the highest risk of HBOC-associated cancers. We also discussed the advantages of utilizing inherited impairment of HR, such as in the drug application for PARPi and platinum chemotherapy, the application of which can be discussed based on the molecular characteristics of HR in malignancies. MGPT is becoming a standard modality to screen possible hereditary tumor-related genes, including HR genes, with a higher potential of comprehensive screening for inherited cancer susceptibilities and an accurate diagnosis of hereditary tumors.

## Figures and Tables

**Figure 1 ijms-23-00348-f001:**
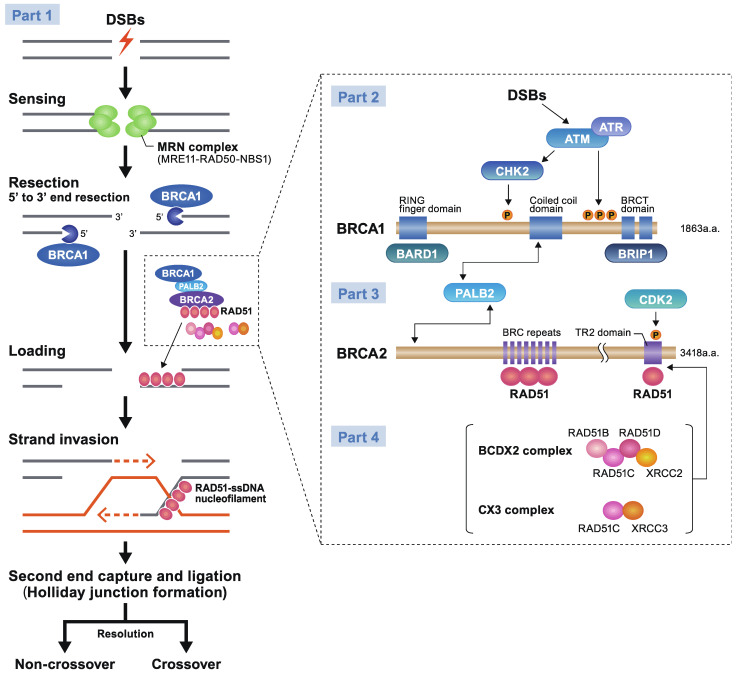
DNA double-strand breaks (DSBs) repair by homologous recombination (HR) and key molecules (MRN complex, ATM, CHK2, BRCA1, PALB2, BARD1, BRIP1, BRCA2, RAD51, RAD51C, and RAD51D). Part 1 (overall HR pathway against DSBs). In response to DSBs or stalled replication fork (not shown), the initial step for HR is the sensing of damaged sites by the MRN complex, which is comprised of MRE11, RAD50, and Nijmegen breakage syndrome protein 1 (NBS1 or, alternatively, NBN), followed by the resection procedure. Either side of DNA is lysed from 5′ to 3′ by MRE11 (shown like “PAC-MAN” in this figure) with its exonuclease and endonuclease activities and BRCA1 enzymatic activities, forming an overhanging 3′ single-stranded DNA (ssDNA) end which can invade homologous template. RAD51 is recruited by the BRCA1–PALB2–BRCA2 effector complex and is loaded on ssDNA to form RAD51–ssDNA nucleofilament. Recombinase RAD51 synthesizes DNA using sister chromatids as a template, extending to the second DNA end, which is terminated by D-loop capture (not shown), and forms a double Holliday junction (not shown), resulting in non-crossover or crossover outcomes via resolution. Parts 2–4 (magnified view of BRCA1–PALB2–BRCA2 complex and molecular associates). In response to DSBs, a serine/threonine kinase ATM, another sensor molecule of DSBs, is recruited with checkpoint activation and phosphorylation of checkpoint kinase 2 (CHK2) to recruit the BRCA complex (Part 2). BRCA1 (1863 amino acids (a.a.)) is phosphorylated by ATM or ATR (ATR works when stalled replication fork occurs) at several motifs containing serine residues. Another serine residue (S988), located in the center of BRCA1, is phosphorylated by CHK2. Phosphorylated BRCA1 is recruited and accumulated at the damaged sites. BRCA1 functions as an adaptor for BRCA2 recruitment through the linking mediator of PALB2. BRCA1 functions by forming a heterodimeric complex with BARD1 via the RING finger domain at the N-terminus of BRCA1 and with BRIP1 through two BRCT domains at the carboxyl-terminus of BRCA1. PALB2 binds to BRCA1 at the coiled-coil domain via the N-terminal residue of PALB2 (Part 3). The N-terminus of BRCA2 interacts with PALB2 through its carboxyl terminus. BRCA2 (3418 amino acids (a.a.)) contains eight conserved motifs called BRC repeats in the middle part encoded by exon 11 to which RAD51s are bound. The carboxyl-terminus of BRCA2 contains a TR2 domain, another binding site for RAD51. BRCA2 serine 3291 (S3,291) phosphorylation, which is required for binding of RAD51 to BRCA2, is regulated by cyclin-dependent kinase 2 (CDK2). RAD51 paralog complexes (RAD51B–RAD51C–RAD51D-XRCC2 (BCDX2) and RAD51C–XRCC3 (CX3)) interact with RAD51 under the regulation of the BRCA1–PALB2–BRCA2 complex and may promote RAD51 nucleofilament formation (Part 4).

**Figure 2 ijms-23-00348-f002:**
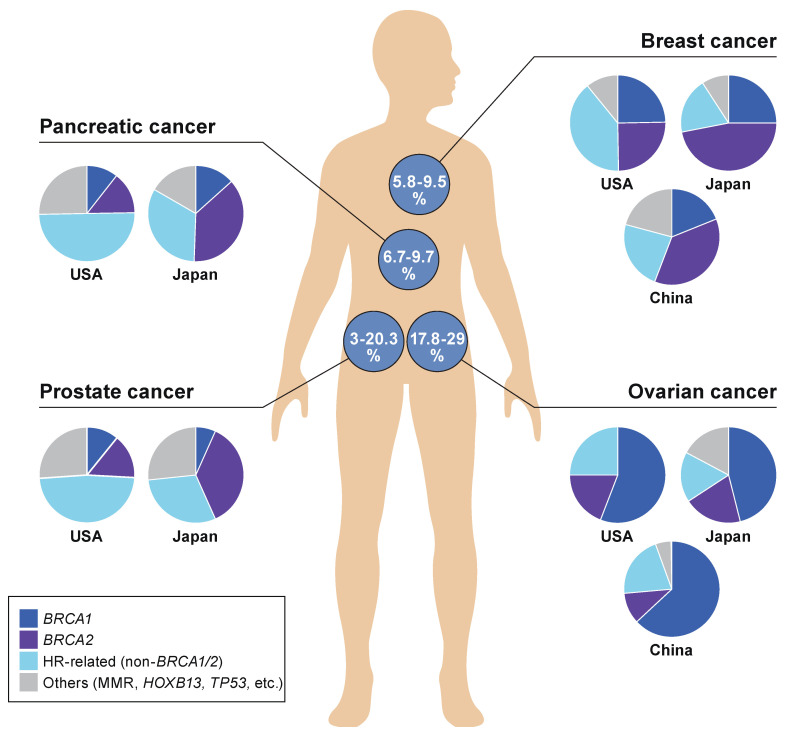
Prevalence of *BRCA* and non-*BRCA* in HRD-related hereditary tumors. The proportion of inheritance backgrounds due to cancer susceptibility gene alterations in each type of malignancy was summed as proportion rates (%), and is shown in circles on the human body figure. The summed rates (%) are based on the data of multi-gene panel testings (MGPTs), performed and analyzed in each country independently [47,52,111,112,113,114,115,116,117,118]. The proportions for the types of each gene alteration are categorized under any of these four groups: *BRCA1*, *BRCA2*, HR-related genes (non-*BRCA1*/*2*), or others (MMR, *HOXB13*, *TP53*, etc.), whose germline variants were determined as pathogenic/deleterious, and are shown in pie charts. Ovarian cancer includes fallopian tube and primary peritoneal carcinomas.

**Table 1 ijms-23-00348-t001:** Representative HR-related genes, functions, and susceptibility to cancer types due to germline variants.

HR-Related Genes		Molecular Functions	Functions in HR	Susceptible Organs	Risk Estimates	References
*ATM*		serine-threonine kinase/checkpoint activation (G2-M and G1-S checkpoints)/p53 activation	BRCA1 phosphorylation	Breast (luminal B-like/HER2-negative), Ovary, Pancreas Prostate, Colon	Breast cancer (2.8–3.3), Ovarian cancer (0.9–2.4), Pancreatic cancer (5.7–9.0), Prostate cancer (2.9), Colorectal cancer (2.8)	[40,52,53,54,55,56,57,58,59]
*BRCA1* (*FANCS*)		mediator/adaptor/enzymatic/cell cycle regulator (G2-M checkpoint control)	BRCT domain-mediated phospho-protein interactions	Breast (TNBC), Ovary (HGSC), Pancreas, Prostate	Breast cancer # (46–87% vs. 12%), Ovarian cancer # (39–63% vs. 1–2%), Pancreatic cancer # (1–3% vs. 0.50%), Prostate cancer # (8.6% by age 65 vs. 6% through age 69)	[54,60,61,62,63,64,65,66,67,68,69,70]
*BARD1*		constitutive RING-mediated heterodimerization with BRCA1	forming BRCA1-BARD1 E3 ubiquitin ligase/interacting with BRCA1 through RING domain	Breast(early-age onset), Ovary	Breast cancer (1.9–3.2), Ovarian cancer (4.2)	[36,40,53,57,59]
*BRCA2* (*FANCD1*)		mediator of RAD51	recombination mediator/RAD51 binding	Breast, Ovary (HGSC), Pancreas, Prostate, Skin (potential risk for Malignant Melanoma)	Breast cancer # (38–84% vs. 12%), Ovarian cancer # (16.5–27% vs. 1–2%), Pancreatic cancer # ( 2–7% vs. 0.50%), Prostate cancer # (15% by age 65; 20% lifetime vs. 6% through age 69)	[54,60,61,62,64,65,66,67,68,69,70]
*BRIP1* (*FANCJ*)		5′ to 3′ DNA helicase/binds to BRCA1/phosphorylated following DNA damage	interacting with BRCA1 through BRCT domain/genome integrity/tumor suppressive	Breast, Ovary, Pancreas, Prostate	Breast cancer (1.6), Ovarian cancer (2.4–8.1), Pancreatic cancer (2.7)	[53,57,58,71]
*CHEK2*		serine/threonine kinase (ATM-mediated)	BRCA1 phosphorylation	Breast (luminal A-like), Prostate, Colon	Breast cancer (2.68), Prostate cancer (ethnic differences in *CHEK2* variant frequencies)	[21,72,73,74]
MRN complex	*MRE11* (*MRE11A*)	enzymatic/nuclease	MRE11, endonuclease; 3′–5′ exonuclease; cooperate with CtIP to initiate DSB resection	Breast	Breast cancer (0.9–9.0)	[40,53,75]
	*RAD50*	structural/ATPase	RAD50, structural maintenance of chromosomes SMC-like protein	–	–	[53,58,59,75]
	*NBS1* (*NBN*)	adaptor/checkpoint	NBS1 (NBN), phospho-protein and ATM kinase interactions	Breast, Ovary, Prostate	Breast cancer (3.2), Ovarian cancer (1.9), Prostate cancer (3.9)	[57,76,77]
*PALB2* (*FANCN*)		scaffold/mediator/partner for BRCA2 stability/nuclear localization	complex formation/linking BRCA1 and BRCA2	Breast, Ovary, Pancreas, Prostate, Colon, Kidney (Wilm’s tumor), CNS (Medulloblastoma)	Breast cancer (2.3–9.5), Ovarian cancer (2.9–4.4), Pancreatic cancer (2.3–14.8), Prostate cancer (0.4), Colorectal cancer (4.9)	[6,53,54,55,56,57,58,59,75,78,79]
*RAD51C*		RAD51 paralog	interacting with RAD51/forming subcomplex (BCDX2 complex)	Breast, Ovary, Pancreas	Breast cancer (0.4–1.4), Ovarian cancer (3.4–5.1)	[40,53,57,58,59,80]
*RAD51D*		RAD51 paralog	interacting with RAD51/forming subcomplex (BCDX2 complex)	Breast, Ovary, Pancreas	Breast cancer (3.1–8.3), Ovarian cancer(4.8–10.9)	[40,53,57,58,59]
*RAD51*		recombinase	final effector of HR/nucleofilament formation/strand invasion	Not specified	–	–
#: Estimated lifetime cumulative risks for the indicated malignancy due to pathogenic germline variants versus general population risks are shown as risk estimates by referring to GeneReviews^®^ [Internet] https://www.ncbi.nlm.nih.gov/books/NBK1247/ (accessed on 19 November 2021) and [81].						

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
