# Peer review of "Homologous Recombination Deficiencies and Hereditary Tumors"

_ijms, 2021, doi:10.3390/ijms23010348_

Round 1

Reviewer 1 Report

The article sent for review ,,Homologous Recombination Deficiencies and Hereditary Tumors” is written in a legible and understandable way. The authors describe the issue in detail and modern. In the opinion of the reviewer, the number of literature items is too long and should be shortened. Additionally, authors should provide a certificate confirming the native speaker correction.

Reviewer 2 Report

Comments to authors

The authors did not pay enough attention to the issue of ethnic specificity of germline mutations in homologous repair genes. At the same time, racial differences are known in the variants of repair genes associated with hereditary breast and ovarian cancer in individuals from different ethnic groups. Such an analysis would be useful and would justify the need to search for germline mutations in the repair genes, which may differ in patients belonging to different races: Caucasian, Mongoloid, and Negroid.

It would be desirable   authors to shortly analyze ethnic specificity  for germline variants of HR genes in different populations.

The manuscript may be accepted  after minor revision taking into account  the ethnic and race differences in germline alterations in  HR related gene    associated with hereditary cancer syndrome.
